# Machine Learning Classification of Event-Related Brain Potentials during a
Visual Go/NoGo Task

**DOI:** 10.3390/e26030220

**Published:** 2024-02-29

**Authors:** Anna Bryniarska, José A. Ramos, Mercedes Fernández

**Affiliations:** 1Department of Computer Science, Opole University of Technology, 45-758 Opole, Poland; a.bryniarska@po.edu.pl; 2College of Computing and Engineering, Nova Southeastern University, Fort Lauderdale, FL 33314, USA; jr1284@nova.edu; 3Department of Psychology and Neuroscience, Nova Southeastern University, Fort Lauderdale, FL 33314, USA

**Keywords:** machine learning, binary classification, EEG signal, state space modeling, biological signal, event-related brain potentials

## Abstract

Machine learning (ML) methods are increasingly being applied to analyze biological
signals. For example, ML methods have been successfully applied to the human
electroencephalogram (EEG) to classify neural signals as pathological or non-pathological
and to predict working memory performance in healthy and psychiatric patients. ML
approaches can quickly process large volumes of data to reveal patterns that may be missed
by humans. This study investigated the accuracy of ML methods at classifying the
brain’s electrical activity to cognitive events, i.e., event-related brain
potentials (ERPs). ERPs are extracted from the ongoing EEG and represent electrical
potentials in response to specific events. ERPs were evoked during a visual Go/NoGo task.
The Go/NoGo task requires a button press on Go trials and response withholding on NoGo
trials. NoGo trials elicit neural activity associated with inhibitory control processes.
We compared the accuracy of six ML algorithms at classifying the ERPs associated with each
trial type. The raw electrical signals were fed to all ML algorithms to build predictive
models. The same raw data were then truncated in length and fitted to multiple dynamic
state space models of order $n_{x}$ using a continuous-time subspace-based system
identification algorithm. The $4n_{x}$ numerator and denominator parameters of the transfer
function of the state space model were then used as substitutes for the data.
Dimensionality reduction simplifies classification, reduces noise, and may ultimately
improve the predictive power of ML models. Our findings revealed that all ML methods
correctly classified the electrical signal associated with each trial type with a high
degree of accuracy, and accuracy remained high after parameterization was applied. We
discuss the models and the usefulness of the parameterization.

## 1. Introduction

There is an increasing interest in the application of machine learning (ML) methods to
analyze biological signals, including signals from the human body [1,2] and electrical signals from the brain, i.e., the electroencephalogram (EEG) and
event-related brain potentials (ERPs) [3,4,5]. ML methods have been successfully applied to EEG
recordings to automate the detection of seizures and improve diagnostic accuracy [6] and to classify emotional states
[7,8]. ML methods have also been successfully applied to
ERPs to improve the diagnostic accuracy and prognosis of attention-deficit hyperactivity
disorder (ADHD) [9]. ERPs are
extracted from the ongoing EEG and represent the sum of electrical potentials that are
time-locked to a cognitive event and are generated by populations of neurons that fire
within milliseconds after the event. The temporal resolution of ERPs is unparalleled by
other brain imaging procedures and they are considered the gold standard for observing
neural activity over time. In [10],
the authors present a comprehensive review of the major techniques used for EEG signal
processing and feature extraction as they relate to decoding and classification of EEG
signals. Other techniques that have been used to capitalize on the information in ERPs are
averaging of the temporal waveforms (i.e., averaged ERPs), time–frequency
representation, and phase dynamics. Indeed, in a recent study [11], the three techniques were applied in combination
with a neural network-based ML model to better exploit the neural dynamic behavior in the
ERP elicited during a visual oddball task. The oddball paradigm requires a response to a
target stimulus that is presented infrequently (e.g., on 20% of trials) within a series of
standard stimuli. The infrequent target stimulus elicits a P3 ERP component. The P3 ERP is a
positive-going wave that occurs 250–500 ms after stimulus onset, with maximum
amplitude over parietal electrode sites, and reflects updating of the memory trace [12]. Results showed that the
three-feature model classified the averaged ERP signal to the rare target and the frequent
standard stimulus with an accuracy level of 86.9%.

ML methods have also been applied to classify neural activity elicited during a Go/NoGo
task. The Go/NoGo task is widely used in cognitive neuroscience to assess frontal-lobe
inhibitory control processes associated with response inhibition and, more generally, with
executive function (EF) [13].
Executive function refers to a set of abilities that work together to regulate thought and
action. The Go/NoGo task requires a button press on Go trials and response withholding on
NoGo trials. The underlying neural marker associated with frontal-lobe inhibitory control
processes is the N2 ERP. The N2 ERP is a negative-going wave in the 200–350 ms
post-stimulus time window, with maximum amplitude over frontal-central electrode sites. NoGo
trials, which require greater inhibitory control, elicit greater N2 ERP amplitude than Go
trials, which require less inhibition [14,15,16]. Indeed, studies of healthy adults
reveal that the amplitude of the N2 ERP is larger in participants who accurately withhold a
response on NoGo trials relative to those who do not withhold a response [17]. One study [18] applied ML methods to identify neural processes of
response selection and response inhibition engaged during the Go and NoGo conditions.
Results revealed an accuracy rate of 92%, estimated by 5-fold cross-validation. Another
study investigated the influence of self-reported personality traits of impulsivity and
compulsivity on performance based on the ERP. Regression tree analyses did not reveal a
relationship between self-reported measures and behavior or the Go/NoGo ERPs [19].

While ML methods have made meaningful contributions to EEG classification, shortcomings
related to EEG data make classification difficult for ML algorithms [20]. For example, ML algorithms have to deal with
signals that are rich in noise. Additionally, most EEG studies involve a small number of
study participants, usually between 10 and 20 [21], permitting only small data sets for the learning
phase of the process. There are two situations that can degrade the performance of ML
algorithms: (1) not having a sufficient number of study participants and (2) having a very
large number of data points. The latter may lead to “the curse of
dimensionality.” For these reasons, it is sometimes difficult to make accurate
classifications of the neural signal, and several techniques must be tested to determine
which ones yield the best results. There are several techniques used to reduce the
dimensionality of EEG data: Linear Discriminant Analysis (LDA), Principal Component Analysis
(PCA), and Independent Component Analysis (ICA) [22]. Discrete Wavelet Transform (DWT) is also often used
for this purpose [23].

We propose a new approach, the use of a state space model as a dimensionality reduction
step, followed by a PCA step to extract the minimum number of significant principal
components (i.e., features) in an optimization approach, coupled with ML. To the knowledge
of the authors, such an approach has not been considered in the literature related to ERP
signals. Notably, state space analysis has been used [24] for estimating multivariate autoregressive (MVAR)
models of cortical connectivity from noisy scalp recorded EEG signals for the purpose of
modeling the spatial covariance structure of the noise in the EEG signal. That study differs
from what we are proposing in that our goal was to substitute the data with parameters and
test the accuracy of ML algorithms at classifying the ERP signals. The rest of the paper is
organized as follows: in Section 2,
we discuss the study methodology and EEG data collection process. In Section 3, we discuss the data reduction process and ML
methodology. In Section 4, we
present the state space methodology for EEG data, while in Section 5 we present the state space analysis. In Section 6, we introduce the system
identification algorithm for impulse response data. In Section 7, we present our results. Finally, in Section 8 we draw conclusions and make
recommendations for future work.

## 2. Study Methodology and EEG Recording

The ERP data reported in this article were collected as part of a larger study
investigating neural and behavioral differences in a linguistically diverse student
population [25]. Participants were
recruited from the main campus of NSU and were invited to participate if they were
right-handed, had normal hearing, normal or corrected-to-normal vision, intact color vision,
met the language requirement, and did not report neurological or psychiatric conditions that
affect cognition.

### 2.1. Participant Information

A total of 268 participants were tested. Data from seven participants were excluded from
the analyses because these participants did not meet study criteria, and six participants
did not yield usable data. Thus, ERP data from 255 study participants were used in the
analyses. Participants were between 18 and 30 years of age (mean =
19.5, SD = 2.73) and the male to female ratio was 55/206.

### 2.2. Visual Go/NoGo Task

The stimuli for the Go/NoGo task were red and green circles, presented on a computer
monitor against a black background, and subtending a visual angle of
$2.9^{{}^{\circ}}$. Each stimulus was presented for 80 ms. Each trial
consisted of two stimuli separated by 1200 ms. For each trial, when a target circle was
followed by another target circle (Go trials), participants pressed a response button to
the second circle. When the target circle was followed by a nontarget circle (NoGo
trials), participants withheld their response. Go and NoGo trials occurred with equal
frequency (36% each trial type). Trials that started with a nontarget stimulus were not
analyzed. The Go/NoGo task consisted of 200 trials, divided into four blocks of 50 trials,
with an intertrial interval (ITI) of 1800 ms. To increase task difficulty, an auditory
signal (300 ms at 1 kHz, 60 dB SPL tone burst) was sounded if the participant did not
respond within 600 ms after the second target stimulus was presented. This time pressure
was introduced after the first 100 trials. Participants focused on a fixation point,
responded as quickly as possible to the second target in the pair on Go trials, and
withheld responding on NoGo trials. The task began after participants read the
instructions on the computer monitor and practiced the task. After the second block of
trials, participants were trained on the task with the added time pressure (tone burst),
after which the remaining two blocks of trials were presented. Participants were
instructed to respond quickly to avoid the tone burst.

### 2.3. EEG Recording and Processing

The continuous EEG was recorded with a lycra cap fitted with 64 Ag/AgCl sintered
electrodes (i.e., 62 scalp electrodes and 2 bipolar electrodes for vertical and horizontal
eye movement recording) and amplified with a Neuvo amplifier (Compumedics U.S.A. Inc.,
Charlotte, NC, USA). The EEG was sampled at 500 Hz, which exceeds the Nyquist frequency
[26]. Eye movement was recorded
with electrodes placed above and below the left eye and on the outer canthus of each eye.
Reference electrodes were placed on the right and left mastoid. Electrode impedance was
maintained at <10 kΩ, and most were under <5 kΩ. After recording, the EEG data were processed offline with
Curry 8 software (Compumedics U.S.A. Inc.). Offline, the EEG was re-referenced to the
common average reference and filtered (high-pass filter set to 0.10 Hz, slope = 0.2; low-pass filter set to 30 Hz, slope =
6.0; 60 Hz notch filter, slope = 1.5). Eyeblinks exceeding ±75μV were corrected using the covariance method [27]. The covariance analysis is
performed between the eye artifact channel and each EEG channel. Linear transmission
coefficients, similar to beta weights, are computed. Based on the weights, a proportion of
the voltage is subtracted from each data point.

Stimulus locked trials (−140 to 800 ms) were then extracted from the ongoing EEG and
baseline (−140 to 0 ms) corrected. The noise statistic was applied to
automatically reject contaminated trials. Noise was computed over the baseline period and
trials that exceeded the average noise level were automatically rejected. Only trials with
correct responses were averaged together by trial type and exported for analysis. Thus,
each participant generated two averaged ERP waves, one Go and one NoGo.

## 3. Data Reduction and Machine Learning Methodology

Our goal was to employ different ML algorithms to show which ones achieve the highest
classification accuracy of the ERP signal as either corresponding to a Go or a NoGo trial.
To achieve this, we divided the data into two sets, both having 510 subjects (255 for the Go
trials and 255 for the NoGo trials) and 62 electrodes. One set of data contained 471 data
points per electrode, i.e., the entire ERP signal, whereas the other set contained only 250
data points per electrode, representing the most significant portion of the ERP signal (see
Figure 1). Due to the fact that
the recorded data has a 3-dimensional (3D) structure, we applied a data unfolding procedure
described in [28] (see Figure 2).

Let *X* denote the data matrix $$\begin{array}{ccc} X & = & \left[\begin{array}{ccccccccccccc} x_{11}^{1} & x_{12}^{1} & \cdots & x_{1T}^{1} & x_{11}^{2} & x_{12}^{2} & \cdots & x_{1T}^{2} & \cdots & x_{11}^{N} & x_{12}^{N} & \cdots & x_{1T}^{N}\\ x_{21}^{1} & x_{22}^{1} & \cdots & x_{2T}^{1} & x_{21}^{2} & x_{22}^{2} & \cdots & x_{2T}^{2} & \cdots & x_{21}^{N} & x_{22}^{N} & \cdots & x_{2T}^{N}\\ \vdots & \vdots & \ddots & \vdots & \vdots & \vdots & \ddots & \vdots & \ddots & \vdots & \vdots & \ddots & \vdots \\ x_{P1}^{1} & x_{P2}^{1} & \cdots & x_{PT}^{1} & x_{P1}^{2} & x_{P2}^{2} & \cdots & x_{PT}^{2} & \cdots & x_{P1}^{N} & x_{P2}^{N} & \cdots & x_{PT}^{N} \end{array}\right], \end{array}$$ where T={250,471} indicates the two different numbers of data points used in
the study, N=62 is the number of electrodes, and P=510 is the number of subjects. For the data set containing 471
data points, *X* would have dimensions 510 × 29,202, which is a fairly
large data set. On the other hand, with only 250 data points, *X* would have
dimensions 510 × 15,500, which is a smaller data set, i.e., a 47% reduction. However, if we could fit dynamic models to the
data set with 250 data points per electrode, then we could use the parameters of the models
as a substitute for the data set. This could be a significant data reduction step, provided
there is no loss of accuracy in modeling the data. One such type of dynamic model comes from
the class of subspace-based state space system identification algorithms, collectively known
as N4SID [29,30,31,32]. The idea would be
to fit 62 state space models to the data containing 250 points per electrode, thus obtaining
a set of 62 parameter triplets {A,B,C}, where $A∼n_{x}\times n_{x}$, $B∼n_{x}\times 1$, and $C∼1\times n_{x}$, thus, totaling $n_{x}^{2}+2n_{x}$ parameters per electrode, where $n_{x}$ is the system order. One could then convert the models to a
transfer function form, which is a more parsimonious representation, resulting in
$2n_{x}$ parameters per electrode, i.e., $n_{x}$ numerator parameters and $n_{x}$ denominator parameters. This could result in a data matrix of
size $510\times 124n_{x}$. Preliminary analyses carried out using data from the entire
data set indicate that using an $n_{x}=20$ results in models with great fidelity. That is, we would
obtain a data matrix of size 510 × 2480, which is much smaller than 510 × 29,202,
by a 91.5% reduction factor. However, the parameters of the transfer
function model could result in being complex numbers, therefore, in the worst case scenario,
one has to split the parameters into their real and imaginary parts, thus accounting for
twice the number of parameters, i.e., 510 × 4960 or an 83% reduction. This approach alleviates the curse of
dimensionality, which is quite common in machine learning. Comparison of the results would
allow for the direct assessment of the effectiveness of dimensionality reduction to EEG
analyses.

ML algorithms create a predictive model based on the provided data: classification labels,
training data, and test data. This is called supervised learning. The available data are
usually divided into training and test or validation data sets. The ML algorithms use the
training data set to build a predictive model, which is then validated with the test data.
Figure 3 shows the overall ML
modeling process. One starts with the training data, along with a set of class labels, i.e.,
{0,1} for binary classification. This information is fed to the ML
algorithm, which in turn uses a K-fold cross-validation procedure to obtain a predictive
model. The test data, which are new to the model, are then used to predict its class labels.
Such models can be employed for classification, much like the ones we use here for
classifying the ERP signal into Go and NoGo trials (thus, a binary classification
problem).

As described above in Section 2,
each of the 255 participants generated an averaged Go and a NoGo ERP signal. Thus, 510 ERP
signals were used in the analyses. Sampled at 500 Hz for 940 ms, (−140 to 800 ms) including a 140 ms pre-stimulus baseline
(−140 to 0 ms), each ERP signal consisted of 471 data points. The
signal was collected from 62 electrodes placed over the scalp of the study participants.
Thus, a matrix containing 29,202 data points (62 electrodes × 471 data points) was
obtained for each participant (see Figure 4). The processed signal was subjected to state space modeling in order to
establish parameters that could replace the entire signal and reduce the dimensionality of
the input data to the ML classifier (see Figure 5). For each electrode, 40 parameters were calculated according to the
state space modeling methodology described in Section 6. Since each parameter is a complex number
containing real and imaginary parts, hence 62 electrodes × 40 parameters × 2 (real
and imaginary) equals 4960 data points after parameterization. For each data sample from a
participant, a reduction in dimensionality by 83% was obtained. These data were then used to perform a PCA to
assess the number of significant principal components as a function of accuracy of the ML
classifier. Six different ML algorithms were analyzed in terms of accuracy versus the number
of significant principal components.

The ML algorithms used in the research are k-nearest neighbors (KNN), Naive Bayes (NB),
decision trees (DTs), linear discriminant analysis (LDA), support vector machines (SVM), and
random forest (RF). KNN is a simple and powerful supervised machine learning algorithm that
can be used for classification tasks. KNN is often used in cases where the data are
nonlinear or do not fit well into traditional parametric models. The NB classifier is a
probabilistic machine learning model based on Bayes’ theorem with an assumption of
independence among predictors. DTs are hierarchical structures used for classification
tasks. They consist of decision nodes that split the data based on features, and leaf nodes,
which represent the outcome. The algorithm selects the best feature to split the data at
each node, aiming to maximize purity. Once constructed, the tree is used to predict outcomes
for new data. Key features include interpretability and the ability to capture complex
decision boundaries. LDA is a statistical model used for topic modeling. It assumes
documents are composed of a mixture of topics, and each topic is characterized by a
distribution of words. LDA aims to identify these topics in a collection of documents. The
SVM method is a supervised learning method that analyzes given data and identifies patterns
which are used for classification and regression analysis [33]. The SVM method is based on the concept of decision
space, which is divided by building boundaries separating objects of different class
affiliation. In binary classification there are two classes, and a boundary line is created
to separate them. This method is widely used to analyze EEG signals of epileptic seizure
activity [34], sleep recordings of
patients [35], and in the
recognition of emotional states [36]. RF builds multiple decision trees during training and outputs the mode for
classification prediction of the individual trees. RF introduces randomness in the
tree-building process by using a subset of features at each split and bootstrapping sample.
This helps in reducing overfitting and improving generalization performance [37].

PCA is a statistical technique used for dimensionality reduction and data visualization.
PCA aims to transform the original data set into a new coordinate system where the variables
(features) are uncorrelated, and the variance along each axis (principal components) is
maximized. This transformation is achieved by identifying the principal components, which
are linear combinations of the original variables [38]. Lastly, the 5-fold cross-validation method is used
to determine the average classification results. The k-fold cross-validation process is
shown in Figure 6, where
k=5. In each fold, different parts of the data set are taken as
the test and training sets. This approach ensures that the outcomes remain unaffected by the
selection of partitioning the data into training and test sets.

## 4. State Space Modeling of EEG Data

In the context of EEG measurements, an impulse response is a signal change that corresponds
to a cerebral response to some stimuli. EEG data are therefore the result of an impulse
response experiment. Thus, EEG data can be modeled as a continuous-time impulse response
state space model of the form (1)$$\begin{array}{ccc} {\dot{x}}_{c}\left(t\right) & = & A_{c}x_{c}\left(t\right)+B_{c}u\left(t\right) \end{array}$$
(2)$$\begin{array}{ccc} y(t) & = & C_{c}x_{c}\left(t\right)+D_{c}u\left(t\right). \end{array}$$

The matrices of parameters are given by (3)$$\begin{array}{ccc} A_{c} & = & \left[\begin{array}{cccc} a_{11} & a_{12} & \cdots & a_{1,n_{x}}\\ a_{21} & a_{22} & \cdots & a_{2,n_{x}}\\ \vdots & \vdots & \ddots & \vdots \\ a_{n_{x},1} & a_{n_{x},1} & \cdots & a_{n_{x},n_{x}} \end{array}\right] \end{array}$$
(4)$$\begin{array}{ccc} B_{c} & = & \left[\begin{array}{c} b_{1}\\ b_{2}\\ \vdots \\ b_{n_{x}} \end{array}\right] \end{array}$$
(5)$$\begin{array}{ccc} C_{c} & = & \left[\begin{array}{cccc} c_{1} & c_{1} & \cdots & c_{n_{x}} \end{array}\right] \end{array}$$
(6)$$\begin{array}{ccc} D_{c} & = & \left[\begin{array}{c} 0 \end{array}\right] \end{array}$$
(7)$$\begin{array}{ccc} x_{c}\left(t\right) & = & \left[\begin{array}{c} x_{1}\left(t\right)\\ x_{2}\left(t\right)\\ \vdots \\ x_{n_{x}}\left(t\right) \end{array}\right]. \end{array}$$ Note that the feedback matrix D=0. Then, in the transfer function form of (Equation 1)–(2), the order of the
numerator polynomial is smaller than the order of the denominator polynomial. In traditional
state space analysis, we have an $n_{x}$-order state space model with respective states, inputs, and
outputs at time *t*, given by $x_{c}\left(t\right)\in {I\;R}^{n_{x}}$, $u\left(t\right)\in {I\;R}^{n_{u}}$, and $y\left(t\right)\in {I\;R}^{n_{y}}$, and $\left\{n_{x},x_{c}\left(0\right),A_{c},B_{c},C_{c},D_{c}\right\}$ are the unknown parameters of the system. Such a model is
known as a multi-input, multi-output (MIMO) state space model. When the input and output
dimensions are scalar values, the model is referred to as a single-input, single-output
(SISO) state space model [39], which
is the case of interest in this study.

The problem we address here is the following: Given a sequence of impulse response data
${\left\{g\left(t\right)\right\}}_{t=0}^{N-1}$, obtained from some experiment, determine the system order
$n_{x}$, initial state vector x(0), and parameters matrices $\left\{A_{c},B_{c},C_{c},D_{c}\right\}$. We can only identify the parameters modulo an invertible
similarity transformation matrix, $T\in {I\;R}^{n_{x}\times n_{x}}$. Therefore, the identified model is not unique. However, the
input/output properties of the model are unique. That is, the Markov parameters
(8)$$\begin{array}{ccc} h(i) & = & \left\{\begin{array}{cc} C_{c}A_{c}^{i-1}B_{c}, & i>0\\ D_{c}, & i=0 \end{array}\right., \end{array}$$ the impulse response parameters (9)$$\begin{array}{ccc} g(t) & = & \left\{\begin{array}{cc} C_{c}e^{A_{c}t}B_{c}, & t\geq 0^{+}\\ D_{c}, & t=0^{-} \end{array}\right., \end{array}$$ and transfer function coefficients (10)H(s)=Cc(sInx−Ac)−1Bc(11)=βnxsnx−1+βnx−1snx−3+⋯+β2s+β1snx+αnxsnx−1+αnx−1snx−2+⋯+α2s+α1 are unique, where $I_{n_{x}}$ is an $n_{x}\times n_{x}$ identity matrix and *s* is the Laplace
variable. The parameters $\left\{\alpha_{1},\alpha_{2},\ldots ,\alpha_{n_{x}},\beta_{1},\beta_{2},\ldots ,\beta_{n_{x}}\right\}$ are the parameters of an observable canonical state space
model of the form $$\begin{array}{ccc} A_{oc} & = & \left[\begin{array}{ccccc} 0 & 0 & \cdots & 0 & -\alpha_{1}\\ 1 & 0 & \cdots & 0 & -\alpha_{2}\\ 0 & 1 & \cdots & 0 & -\alpha_{3}\\ \vdots & \vdots & \ddots & \vdots & \vdots \\ 0 & 0 & \cdots & 0 & -\alpha_{n_{x}-1}\\ 0 & 0 & \cdots & 1 & -\alpha_{n_{x}} \end{array}\right]\\ B_{oc} & = & \left[\begin{array}{c} \beta_{1}\\ \beta_{2}\\ \vdots \\ \beta_{n_{x}} \end{array}\right]\\ C_{oc} & = & \left[\begin{array}{cccc} 0 & 0 & \cdots & 1 \end{array}\right]\\ D_{oc} & = & \left[\begin{array}{c} 0 \end{array}\right]. \end{array}$$ Therefore, the minimum number of parameters needed to represent
the state space system (Equation 1)–(2) is $2n_{x}$, if the initial states are ignored. There is a similarity
transformation matrix, $T\in {I\;R}^{n_{x}\times n_{x}}$, such that $A_{oc}=TA_{c}T^{-1}$, $B_{oc}=TB_{c}$, and $C_{oc}=C_{c}T^{-1}$. Note that y(t)=g(t) when u(t)=δ(t), the Dirac delta function. To identify the continuous-time
model, we use the impulse response coefficients and apply Kung’s realization
algorithm [29] to determine
$\left\{n_{x},x_{c}\left(0\right),A_{c},B_{c},C_{c}\right\}$ directly from the data. In Section 5, we will use this approach.

## 5. Identification of $\left\{n_{x},A_{c},B_{c},C_{c}\right\}$ via the Impulse Response Coefficients

One can identify the continuous-time model (Equation 1)–(2) using the measured impulse response
coefficients and Equation (Equation 9). This leads to a Hankel matrix decomposition of the form $$\begin{array}{ccc} G & = & \left[\begin{array}{ccccc} g(0) & g(1) & g(2) & \cdots & g(j-1)\\ g(1) & g(2) & g(3) & \cdots & g(j)\\ g(2) & g(3) & g(4) & \cdots & g(j+1)\\ \vdots & \vdots & \vdots & \ddots & \vdots \\ g(i-1) & g(i) & g(i+1) & \cdots & g(N-1) \end{array}\right]. \end{array}$$ Note that the Hankel matrix is characterized by having constant
antidiagonals. The matrix *G* needs to be factored into the product of the
observability $\left(O_{c}\right)$ and controllability $\left(C_{c}\right)$ matrices, two rank $n_{x}$ matrices. Such matrix decomposition is possible via the
singular value decomposition (SVD) [29,30,31,32]. That is, $$\begin{array}{ccc} G & = & \left[\begin{array}{ccccc} C_{c}B_{c} & C_{c}e^{A_{c}\Delta T}B_{c} & C_{c}e^{2A_{c}\Delta T}B_{c} & \cdots & C_{c}e^{\left(j-1\right)A_{c}\Delta T}B_{c}\\ C_{c}e^{A_{c}\Delta T}B_{c} & C_{c}e^{2A_{c}\Delta T}B_{c} & C_{c}e^{3A_{c}\Delta T}B_{c} & \cdots & C_{c}e^{jA_{c}\Delta T}B_{c}\\ C_{c}e^{2A_{c}\Delta T}B_{c} & C_{c}e^{3A_{c}\Delta T}B_{c} & C_{c}e^{4A_{c}\Delta T}B_{c} & \cdots & C_{c}e^{\left(j+1\right)A_{c}\Delta T}B_{c}\\ \vdots & \vdots & \vdots & \ddots & \vdots \\ C_{c}e^{\left(i-1\right)A_{c}\Delta T}B_{c} & C_{c}e^{iA_{c}\Delta T}B_{c} & C_{c}e^{\left(i+1\right)A_{c}\Delta T}B_{c} & \cdots & C_{c}e^{\left(N-1\right)A_{c}\Delta T}B_{c} \end{array}\right]\\ & = & \left[\begin{array}{c} C_{c}\\ C_{c}e^{A_{c}\Delta T}\\ C_{c}e^{2A_{c}\Delta T}\\ \vdots \\ C_{c}e^{\left(i-1\right)A_{c}\Delta T} \end{array}\right]\cdot \left[\begin{array}{ccccc} B_{c} & e^{A_{c}\Delta T}B_{c} & e^{2A_{c}\Delta T}B_{c} & \cdots & e^{\left(j-1\right)A_{c}\Delta T}B_{c} \end{array}\right]\\ & = & \left[\begin{array}{c} C_{c}\\ C_{c}\left(e^{A_{c}\Delta T}\right)\\ C_{c}{\left(e^{A_{c}\Delta T}\right)}^{2}\\ \vdots \\ C_{c}{\left(e^{A_{c}\Delta T}\right)}^{i-1} \end{array}\right]\cdot \left[\begin{array}{ccccc} B_{c} & \left(e^{A_{c}\Delta T}\right)B_{c} & {\left(e^{A_{c}\Delta T}\right)}^{2}B_{c} & \cdots & {\left(e^{A_{c}\Delta T}\right)}^{j-1}B_{c} \end{array}\right]\\ & = & \left[\begin{array}{cc} U_{1} & U_{2} \end{array}\right]\left[\begin{array}{cc} S_{1} & 0_{n_{x}\times \left(j-n_{x}\right)}\\ 0_{\left(i-n_{x}\right)\times n_{x}} & 0_{\left(i-n_{x}\right)\times \left(j-n_{x}\right)} \end{array}\right]\left[\begin{array}{c} V_{1}^{T}\\ V_{2}^{T} \end{array}\right]\\ & = & U_{1}S_{1}V_{1}^{T}, \end{array}$$ where $U=\left[\begin{array}{cc} U_{1} & U_{2} \end{array}\right]$ and $V=\left[\begin{array}{cc} V_{1} & V_{2} \end{array}\right]$ are orthogonal matrices, and $$\begin{array}{ccc} S_{1} & = & \left[\begin{array}{cccc} s_{1} & & & \\ & s_{2} & & \\ & & \ddots & \\ & & & s_{n_{x}} \end{array}\right] \end{array}$$ is a diagonal matrix of the $n_{x}$ most significant singular values of the continuous-time
system (Equation 1)–(2), thus
$n_{x}$ is the best estimate of the system order. From the above
matrix decomposition we can compute the observability and controllability matrices,
$O_{c}$ and $C_{c}$, respectively, from $$\begin{array}{ccc} O_{c} & = & U_{1}S^{\frac{1}{2}}\\ C_{c} & = & S^{\frac{1}{2}}V_{1}^{T}. \end{array}$$ Furthermore, we need to define two shifted observability
matrices $O_{c}^{1}$ and $O_{c}^{2}$ as $$\begin{array}{ccc} O_{c}^{1} & = & \left[\begin{array}{c} C_{c}\\ C_{c}\left(e^{A_{c}\Delta T}\right)\\ C_{c}{\left(e^{A_{c}\Delta T}\right)}^{2}\\ \vdots \\ C_{c}{\left(e^{A_{c}\Delta T}\right)}^{i-2}, \end{array}\right]\\ O_{c}^{2} & = & \left[\begin{array}{c} C_{c}\left(e^{A_{c}\Delta T}\right)\\ C_{c}{\left(e^{A_{c}\Delta T}\right)}^{2}\\ \vdots \\ C_{c}{\left(e^{A_{c}\Delta T}\right)}^{i-1} \end{array}\right]. \end{array}$$ Likewise, we define the matrix exponential
$e^{A\Delta T}$ as $$\begin{array}{ccc} e^{A\Delta T} & = & {\left(O_{c}^{1}\right)}^{†}O_{c}^{2}, \end{array}$$ where ΔT is the sampling interval.

We can now identify the parameters $\left\{n_{x},A_{c},B_{c},C_{c}\right\}$ from $$\begin{array}{ccc} A_{c} & = & \frac{\log_{e}\left({\left(O_{c}^{1}\right)}^{†}O_{c}^{2}\right)}{\Delta T}\\ B_{c} & = & C_{c}\left(:,1:n_{u}\right)\\ C_{c} & = & O_{c}\left(1:n_{y},:\right)\\ n_{x} & = & \mathrm{rank}\{G\}, \end{array}$$ where $\log_{e}\left(M\right)$ is the base *e* matrix logarithm of the matrix
*M* [31].

## 6. System Identification of an EEG Signal

Here, we have taken EEG data from a single electrode and conducted a system identification
exercise on the data. Figure 7 shows
the singular values versus system order plot, showing a significant cut-off at around
$n_{x}=$ 17. Also evident is the noise floor of singular values for
$\sigma_{x}>$ 22. The fitting error was calculated as
$$\begin{array}{c} f=\left(\frac{1}{N}\right)\sum_{t=0}^{N-1}{\left(g\left(t\right)-g_{fitted}\left(t\right)\right)}^{2}, \end{array}$$ where g(t) is the observed impulse response (observed EEG data),
$g_{fitted}\left(t\right)$ is the fitted impulse response (fitted EEG data), and
N=250 is the number of observations. The fitting error was
$f=3.3271\times 10^{-7}$. Clearly, a state space model with $n_{x}=20$ performed very well, as can be seen in Figure 8.

It is clear that the singular value plot cuts off between $17<n_{x}<21$. Several tests showed that not all electrodes had the same
system order properties as the example above. Therefore, we set the system order to
$n_{x}=20$ for all the models. We selected electrode 19 as an example
and fitted a state space model to it. The system order was between 17 and 21. We chose
n=20 and the mean squared error (MSE) was in the order of
$10^{-7}$. Not all electrodes showed a constant system order
throughout. However, the average system order was about 20. For each electrode, we conducted
an optimization of system order versus MSE. We decided to take n=20 as an average system order and either truncate or zero pad
the transfer function parameters accordingly. This was a result of the decaying behavior of
the parameters in the transfer function as the system order increased. So, we started with a
minimum system order of n=17 and calculated the MSE. We then increased the system order to
n=18 and calculated the new MSE. If the new MSE improved, we kept
increasing the system order by one, thus trying to bound the MSE to a minimum. In essence,
we obtained the optimal system order for each electrode. Since we computed the transfer
function parameters, we either truncated the parameters to n=20 or zero padded them in cases where n<20. The singular value plot versus system order is a common tool
used in state space modeling for determining the system order [29,30].

## 7. Results

First, all six ML models (KNN, NB, DT, LDA, SVM, RF) were tested on the full data set
before applying state space modeling for dimensionality reduction. Matrices as large as 510
× 29,202 data points were taken into account for each study participant. For each ML
algorithm, accuracy results are presented as a function of the number of principal
components required to achieve the given accuracy. The PCA method was used to calculate the
score matrix, and a given subset of principal components were used in a 5-fold
cross-validation analysis for each ML method. The results are presented in Table 1. The best results for each ML
model are marked in blue font for ease of readability.

Table 2 shows the best results
for each of the ML algorithms from Table 1 with calculated metrics [40]:(12)$$\begin{array}{ccc} ERR & = & \frac{FP+FN}{FP+FN+TP+TN} \end{array}$$(13)$$\begin{array}{ccc} ACC & = & \frac{TP+TN}{FP+FN+TP+TN}\;=\;1-ERR \end{array}$$(14)$$\begin{array}{ccc} SPE & = & \frac{TN}{FP+TN} \end{array}$$(15)$$\begin{array}{ccc} SEN & = & \frac{TP}{FN+TP} \end{array}$$(16)$$\begin{array}{ccc} PRE & = & \frac{TP}{FP+TP} \end{array}$$(17)$$\begin{array}{ccc} F1 & = & 2\times \frac{PRE\times SEN}{PRE+SEN}, \end{array}$$ where ERR denotes the error, ACC denotes accuracy, SPE denotes
specificity, SEN denotes sensitivity, PRE denotes precision, and a measure of model
performance is the F1 statistic. Accuracy is a widely used metric for evaluating
classification models, representing the proportion of correctly classified samples among the
total samples assessed. Precision, on the other hand, calculates the ratio of accurately
predicted positive cases to the sum of all positively predicted cases, where TP represents
the true positives and FP represents the false positives, thus precision reveals the
accuracy of positive predictions. Sensitivity, also known as recall or true positive rate,
determines the ratio of TP to the sum of false negatives (FNs) and TPs, thus it highlights
the model’s capability in correctly identifying actual positive cases. Specificity
can be described as the model’s ability to predict a true negative (TN) of each
category available. In the literature, it is also known as the true negative rate. The F1
metric combines both precision and recall to provide a single score that balances the
trade-off between them. Thus, the F1 statistic uses the average measures of sensitivity and
precision to calculate the F-score statistic. It is calculated as the harmonic mean of
precision and recall. It is particularly useful when there is an imbalance between the
classes in the data set. The metrics {ACC, ERR, PRE, SEN, SPE, F1} were used as measures of
fidelity toward judging the performances of the different models. Note that all metrics are
scalars in the range [0,1], with higher values indicating a better model performance, except
for the error metric, ERR, in which a lower value indicates a better model performance since
it is 1−ACC. See Figure 9 for the confusion matrix as a function of TP, TN, FP, and FN.

After using the state space modeling procedure on the raw data matrix *X* of
size 510 × 15,500, thus resulting in a reduced data matrix of size 510 × 4960, it
was then fed to the same six ML algorithms {KNN, NB, DT, LDA, SVM, RF} versus PCA and using
5-fold cross-validation. Once again, accuracy results are presented as a function of the
number of principal components required to achieve the given accuracy. The results are
presented in Table 3.

Shown in Table 4 are the best
results from Table 3 with
calculated metrics: {ACC, ERR, PRE, SEN, SPE, F1}.

We varied the neighboring parameter *k* and number of principal components
as a function of accuracy for the KNN models. The results are shown in Figure 10 for the full data set and in Figure 11 for the reduced data set. As
can be seen, only one neighbor and one principal component were required for accuracies of
96% and 97%, respectively. Based on the overall results, it can be
concluded that ML algorithms showed similarly high accuracy despite a much smaller number of
input data after parameterization. Only in the case of the LDA model, can a reduction in the
effectiveness of the model be observed. In the case of the remaining ML methods, there is
not even a slight change in the results. This means that the use of state space modeling
does not affect the accuracy of ML models and additionally allows for obtaining similar
results to the case of using the full data set. It should be emphasized that state space
modeling reduced the dimensionality by 83%.

## 8. Conclusions

We applied six different ML algorithms to analyze and classify EEG signals collected from
62 scalp electrodes, and we used state space modeling to reduce dimensionality before
applying these algorithms. Our findings revealed that the algorithms yielded high accuracy
rates comparable to those obtained without application of the state space modeling. The
obtained results are important because the use of state space modeling for this purpose has
not been previously described in the literature and may spark new ideas for the development
of ML algorithms.

It is worth noting that, when working with large data sets, dimensionality reduction is
essential for signal classification, noise reduction, and may ultimately improve the
predictive power of ML models. Furthermore, it is important to weigh the trade-offs between
size of the data matrices and the number of parameters, where a parsimonious model (i.e., a
model with a minimum number of parameters) is always preferred.

The ML methods employed in this study successfully classified, with a high degree of
accuracy, Go and NoGo trials in a task in which Go and NoGo trials were equiprobable, which
made it more difficult to distinguish between the two trial types. Go trials are usually
presented more often than NoGo trials, e.g., 80%/20%, respectively, which primes the Go response. Once primed,
greater control is required to stop or inhibit the Go response during NoGo trials. We
presented an equal number of Go and NoGo trials because when an unequal number is presented,
it cannot be determined whether the neural response on NoGo trials is due to response
inhibition or to the relative novelty of the less frequent NoGo stimulus [15,16]. Thus, to avoid the influence of stimulus
probability, we presented Go and NoGo trials with equal frequency. Research shows that when
Go and NoGo trials occur with equal frequency, the neural response to the Go and NoGo trials
is more similar, which increases the difficulty of distinguishing between trial types [14,41]. Our findings suggest that ML algorithms may be
useful to classify neural electrical responses that may otherwise be difficult to
distinguish. For instance, in early or pre-clinical cases associated with deficient
inhibition, such as ADHD and Parkinson’s Disease, ML algorithms may assist with early
detection and diagnosis since research reveals smaller NoGo N2 ERP amplitude in patients
compared to controls [42,43]. In pre-clinical cases, ML
algorithms may detect small changes in the N2 ERP signal that may be missed by visual
inspection alone.

Compared to existing methods, the use of state space modeling on preprocessed data used in
ML algorithms makes it possible to reduce the sizes of the input data. This allows ML
algorithms to run faster and to use a larger number of input variables to classify data,
even with a small number of samples. Reducing dimensionality also significantly affects the
running time of ML algorithms. This approach is important because a smaller number of input
parameters has a positive impact on the interpretability of the results and the operation of
ML algorithms that are susceptible to overfitting. Given the successful application of state
space modeling to ERP signals in the current study, future studies may want to explore this
data reduction approach in other biological signals.

## Figures and Tables

**Figure 1 entropy-26-00220-f001:**
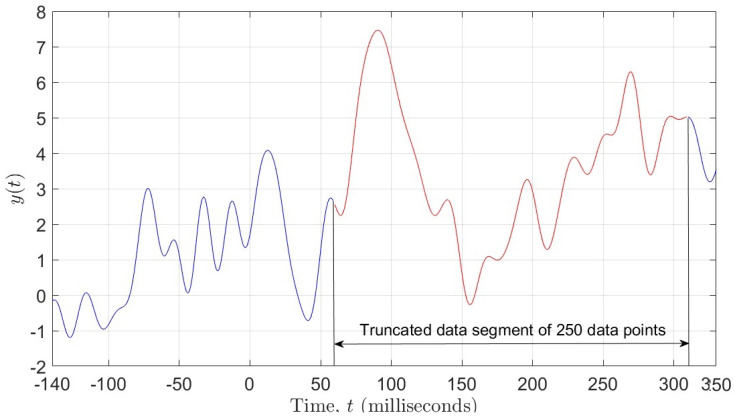
Figure showing 250 data points selected from the ERP signal.

**Figure 2 entropy-26-00220-f002:**
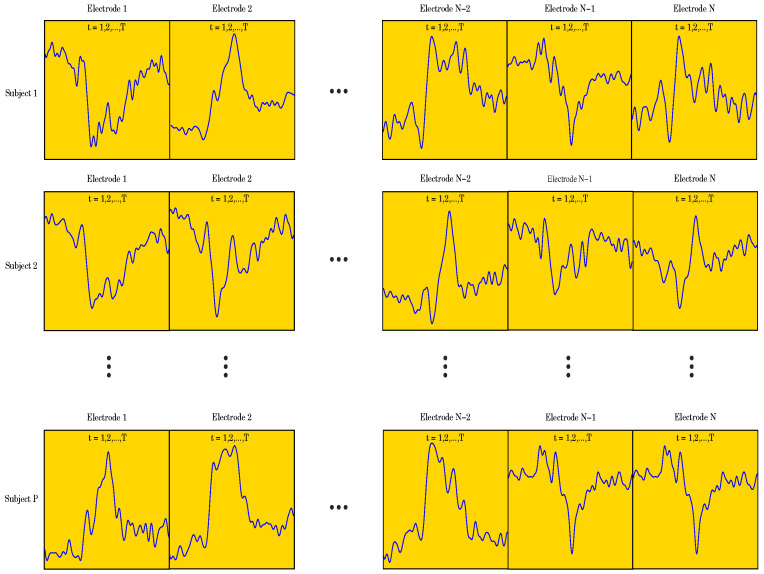
Figure showing the 3D structure of the data for the case when T=250.

**Figure 3 entropy-26-00220-f003:**
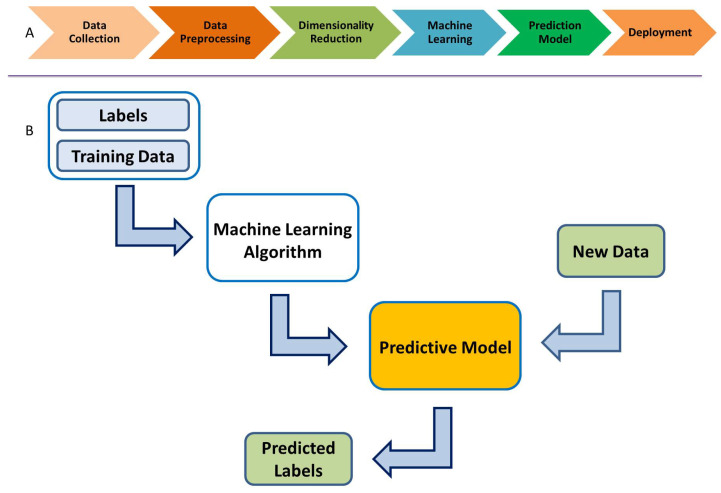
(**A**) Supervised machine learning process. (**B**) Predictive
supervised machine learning.

**Figure 4 entropy-26-00220-f004:**
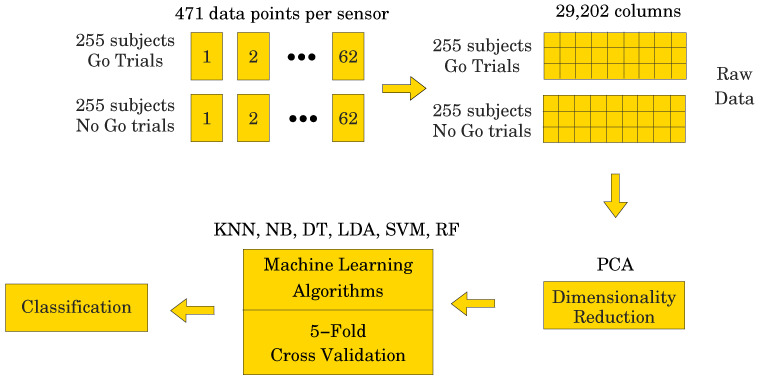
Figure showing the raw data unfolding process.

**Figure 5 entropy-26-00220-f005:**
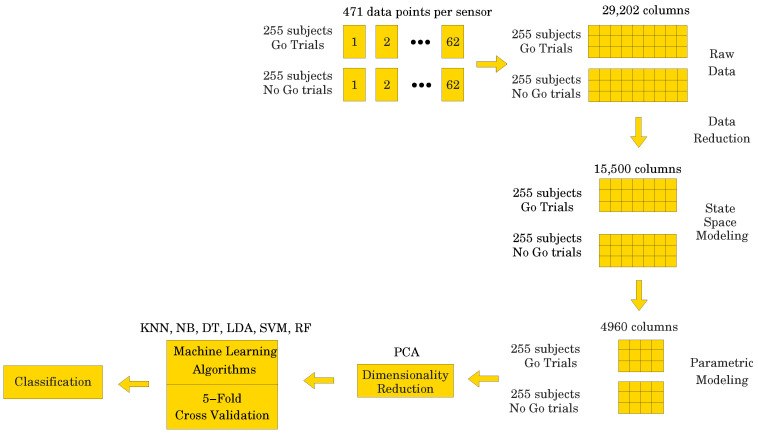
Figure showing the parametric data unfolding process.

**Figure 6 entropy-26-00220-f006:**
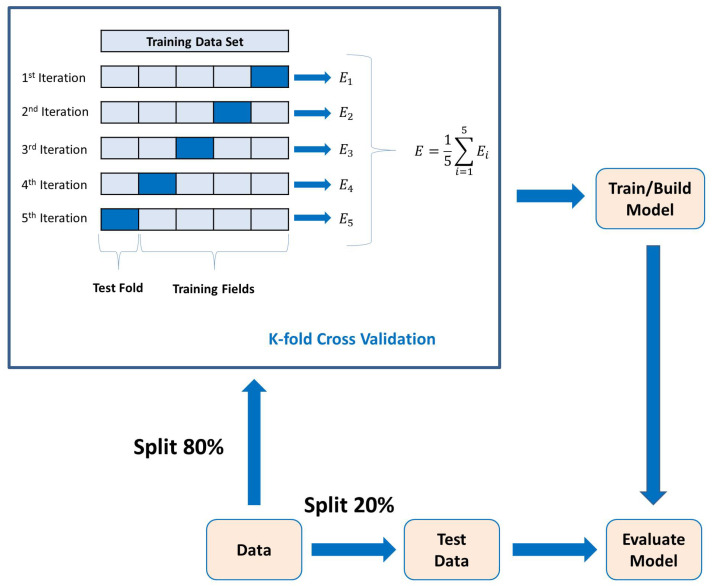
K-fold cross-validation process with model evaluation.

**Figure 7 entropy-26-00220-f007:**
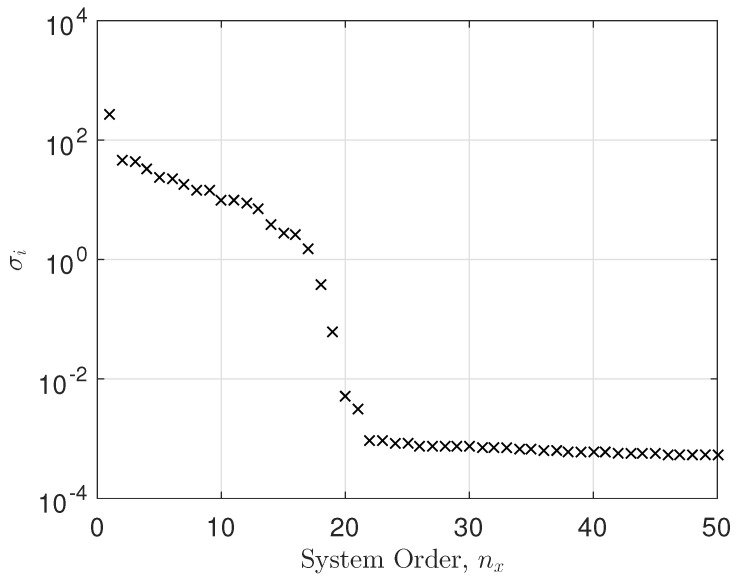
Singular value plot.

**Figure 8 entropy-26-00220-f008:**
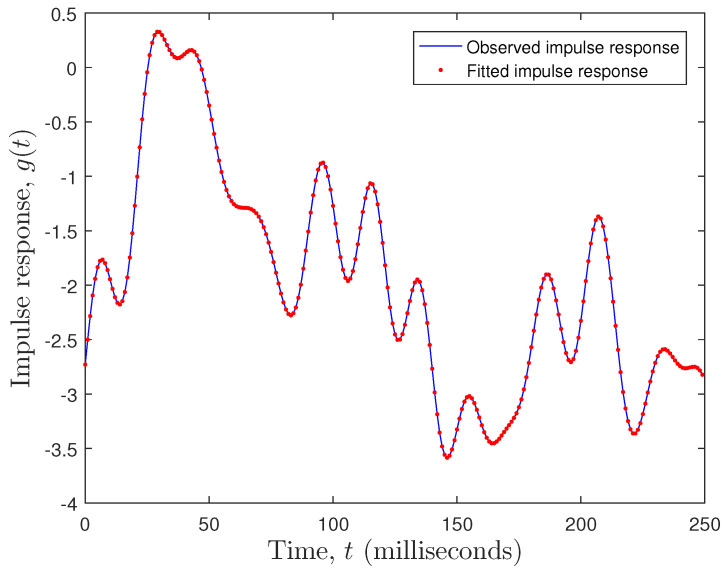
Observed versus fitted EEG plot.

**Figure 9 entropy-26-00220-f009:**
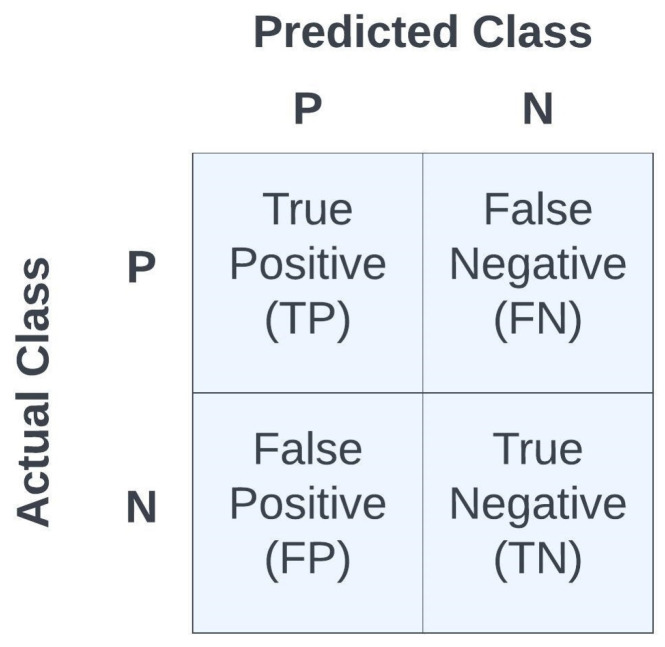
Summary of confusion matrix terminologyfor binary classification.

**Figure 10 entropy-26-00220-f010:**
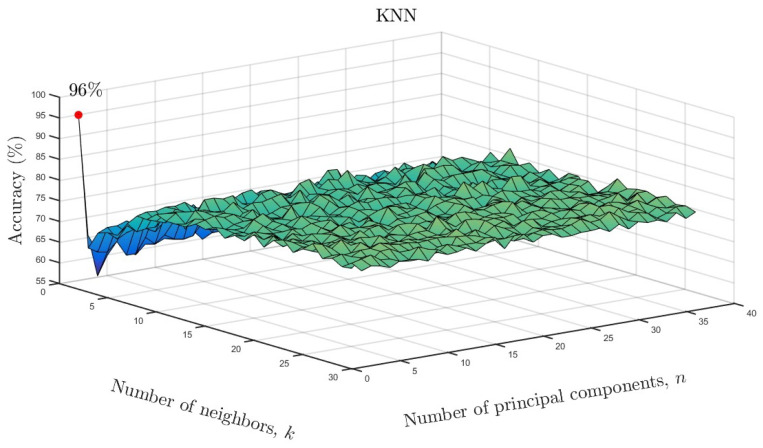
KNN accuracy as a function of number of neighbors and number of significant principal
components for the full data set.

**Figure 11 entropy-26-00220-f011:**
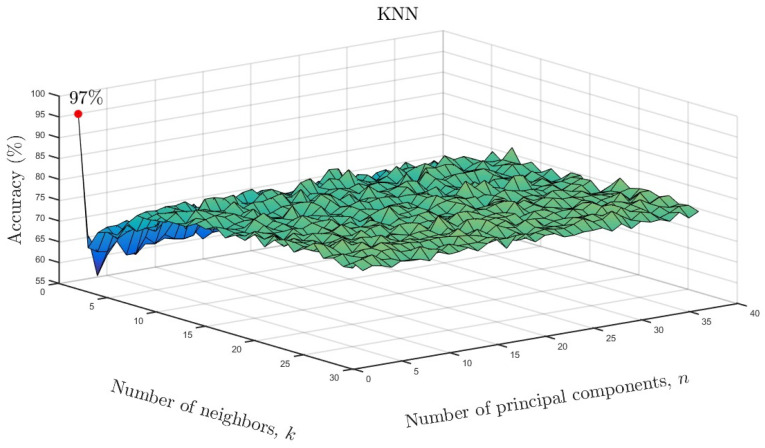
KNN accuracy as a function of number of neighbors and number of significant principal
components for the reduced data set.

**Table 1 entropy-26-00220-t001:** Machine learning algorithm performance versus number of principal components for the
entire ERP signal, i.e., X ∼ 510 × 29,202.

PC (n)	KNN	NB	DT	LDA	SVM	RF
1	0.9627	0.8902	1.0000	0.6098	0.9725	1.0000
2	0.6647	0.8549	1.0000	0.6863	0.6392	1.0000
3	0.5627	0.8510	1.0000	0.6608	0.6294	1.0000
4	0.6078	0.8588	1.0000	0.6725	0.6196	1.0000
5	0.6412	0.8529	1.0000	0.7098	0.6588	1.0000
6	0.6118	0.8314	1.0000	0.7157	0.6529	1.0000
7	0.5980	0.8176	1.0000	0.7157	0.6863	1.0000
8	0.6196	0.8647	1.0000	0.7784	0.7137	1.0000
9	0.6157	0.8431	1.0000	0.7843	0.7000	1.0000
10	0.6255	0.8667	1.0000	0.8059	0.7216	1.0000
11	0.6196	0.8667	1.0000	0.8157	0.7588	1.0000
12	0.6157	0.8706	1.0000	0.8157	0.7431	1.0000
12	0.6333	0.8941	1.0000	0.8510	0.7882	1.0000
14	0.6078	0.8529	1.0000	0.8490	0.7765	1.0000
15	0.6196	0.8745	1.0000	0.8588	0.7961	1.0000
16	0.6549	0.9039	1.0000	0.9020	0.8333	1.0000
17	0.6176	0.9000	1.0000	0.8980	0.8431	1.0000
18	0.6471	0.9137	1.0000	0.8961	0.8333	1.0000
19	0.6196	0.9000	1.0000	0.9078	0.8451	1.0000
20	0.6176	0.9235	1.0000	0.9392	0.8549	1.0000
21	0.6078	0.8980	1.0000	0.9392	0.8627	1.0000
22	0.6275	0.9039	1.0000	0.9392	0.8627	1.0000
23	0.6176	0.8941	1.0000	0.9392	0.8745	1.0000
24	0.6176	0.8941	1.0000	0.9373	0.8706	1.0000
25	0.6137	0.8882	1.0000	0.9392	0.8725	1.0000
26	0.6373	0.8745	1.0000	0.9431	0.8765	1.0000
27	0.6294	0.8706	1.0000	0.9373	0.8686	1.0000
28	0.6059	0.8863	1.0000	0.9412	0.8941	1.0000
29	0.6412	0.8569	1.0000	0.9431	0.8686	1.0000
30	0.6294	0.8922	1.0000	0.9392	0.8588	1.0000
31	0.6078	0.8745	1.0000	0.9431	0.8745	1.0000
32	0.6059	0.8627	1.0000	0.9412	0.8882	1.0000
33	0.6020	0.8725	1.0000	0.9353	0.8882	1.0000
34	0.5941	0.8647	1.0000	0.9353	0.8706	1.0000
35	0.6098	0.8627	1.0000	0.9275	0.8824	1.0000
36	0.6039	0.8490	1.0000	0.9333	0.8863	1.0000

**Table 2 entropy-26-00220-t002:** Summary of machine learning algorithm performances for the entire ERP signal, i.e., X
∼ 510 × 29,202.

Metrics	KNN	NB	DT	LDA	SVM	RF
ACC	0.9627	0.9235	1.0000	0.9431	0.9725	1.0000
ERR	0.0373	0.0765	0.0000	0.0608	0.0275	0.0000
PRE	0.9629	0.9241	1.0000	0.9392	0.9726	1.0000
SEN	0.9627	0.9235	1.0000	0.9392	0.9725	1.0000
SPE	0.9627	0.9235	1.0000	0.9392	0.9725	1.0000
F1	0.9627	0.9235	1.0000	0.9392	0.9725	1.0000

**Table 3 entropy-26-00220-t003:** Machine learning algorithm performance versus number of principal components for the
parametric data case, i.e., 510 × 4960 data points.

PC(n)	KNN	NB	DT	LDA	SVM	RF
1	0.9706	0.9333	1.0000	0.4902	0.9765	1.0000
2	0.7647	0.7843	1.0000	0.5784	0.7902	1.0000
3	0.5824	0.7549	1.0000	0.5941	0.6059	1.0000
4	0.5961	0.7431	1.0000	0.6020	0.6255	1.0000
5	0.5824	0.7549	1.0000	0.6098	0.6373	1.0000
6	0.5706	0.7529	1.0000	0.6235	0.6431	1.0000
7	0.6020	0.7451	1.0000	0.6255	0.6392	1.0000
8	0.5745	0.7471	1.0000	0.6176	0.6373	1.0000
9	0.5882	0.7412	1.0000	0.6196	0.6294	1.0000
10	0.5804	0.7078	1.0000	0.6020	0.6373	1.0000
11	0.5686	0.7196	1.0000	0.6353	0.6588	1.0000
12	0.5784	0.7059	1.0000	0.6333	0.6373	1.0000
12	0.5608	0.7000	1.0000	0.6294	0.6490	1.0000
14	0.5922	0.6784	1.0000	0.6196	0.6490	1.0000
15	0.6059	0.7059	1.0000	0.6235	0.6392	1.0000
16	0.5922	0.6804	1.0000	0.6176	0.6529	1.0000
17	0.5765	0.6667	1.0000	0.6275	0.6490	1.0000
18	0.5529	0.6804	1.0000	0.6255	0.6294	1.0000
19	0.5725	0.6529	1.0000	0.6235	0.6549	1.0000
20	0.5843	0.6745	1.0000	0.6255	0.6608	1.0000
21	0.5549	0.6588	1.0000	0.6333	0.6373	1.0000
22	0.5784	0.6196	1.0000	0.6353	0.6490	1.0000
23	0.5882	0.6294	1.0000	0.6353	0.6431	1.0000
24	0.6039	0.6549	1.0000	0.6569	0.6529	1.0000
25	0.5863	0.6451	1.0000	0.6608	0.6667	1.0000
26	0.5608	0.6431	1.0000	0.6667	0.6510	1.0000
27	0.5784	0.6333	1.0000	0.6647	0.6608	1.0000
28	0.5686	0.6471	1.0000	0.6725	0.6588	0.9980
29	0.5471	0.6412	1.0000	0.6706	0.6647	1.0000
30	0.5824	0.6255	1.0000	0.6647	0.6569	1.0000
31	0.5686	0.6275	1.0000	0.6647	0.6765	1.0000
32	0.5961	0.6294	1.0000	0.6745	0.6765	1.0000
33	0.5941	0.6353	1.0000	0.7078	0.6627	0.9961
34	0.6333	0.6451	1.0000	0.7157	0.6824	1.0000
35	0.5784	0.6471	1.0000	0.7235	0.6804	0.9980
36	0.6098	0.6451	1.0000	0.7216	0.6510	1.0000

**Table 4 entropy-26-00220-t004:** Summary of machine learning algorithm performances parametric data set, i.e., 510
× 4960 data points.

Metrics	KNN	NB	DT	LDA	SVM	RF
ACC	0.9706	0.9333	1.0000	0.7235	0.9765	1.0000
ERR	0.0294	0.0667	0.0000	0.2843	0.0235	0.0000
PRE	0.9706	0.9340	1.0000	0.7288	0.9772	1.0000
SEN	0.9706	0.9333	1.0000	0.7157	0.9765	1.0000
SPE	0.9706	0.9333	1.0000	0.7157	0.9765	1.0000
F1	0.9706	0.9333	1.0000	0.7116	0.9765	1.0000

## Data Availability

The dataset used in this study will be made available upon request.

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

Parkinson’s disease revealed by magnetoencephalographic recording. Sci. Rep..

