# Peer review of "Machine Learning Classification of Event-Related Brain Potentials during a Visual Go/NoGo Task"

_entropy, 2024, doi:10.3390/e26030220_

Round 1

Reviewer 1 Report

Comments and Suggestions for Authors

The manuscript by Bryniarska et al., is about the use of state space model method as feature transformation step in a machine learning-based application in an ERP classification task.

Below I reported my main criticisms:

1.       In results section authors compared their models considering parametrization (does it means after state space model step?) and without parametrization using 4 derivations. Why did they not present the results regarding 62 electrodes without parametrization? It is a crucial point to assess the quality of the models and methods. Otherwise, it is not clear to me how the authors can report the following statement “Without parameterization, the algorithms had difficulty withsignal classification when all 62 scalp electrodes were used in the model.” I’m referring mainly to Table 1 and its related discussion.

2.       Moreover, it seems that the models performed better without parametrization. Therefore, did it mean that the state space model is inefficient or not helpful? In other words, why anyone should consider using this approach if it is not able to improve performance?

3.       Introduction is too short, background and previous studies regarding ERP analysis and machine-learning analysis in Go/No task have been poorly explained. Moreover, the rationale of the study is not well presented.Authors should better clarify why it could necessary the use of machine-learning approaches in this field, as well as why state space model should be preferred to other techniques.

4.       Please provide the IRB approval code.

5.       Was enough the latency of 600ms to discriminate the ERP components from the Go/NoGo task and the ones from the auditory stimulus? Also considering the sampling frequency used. Authors should add some references to justify this choice.

6.       “Eyeblinks exceeding ±75 µV were corrected using the covariance method.” Please add a reference and further details regarding the use of this artifact correction method.

7.       Regarding the demographic information, did the authors consider the gender as confounding factor in their study?

8.       “Here we have taken EEG data from a single electrode and conducted a system identification exercise on the data.” Which electrode did the authors consider? Did the parameters change according to the electrode selected?” “Several testsshowed that not all electrodes had the same system order properties as the example above.” How many tests did the author performed? Did they have any supplementary materials to justify their choice of the system order?

9.       Why did author not compare their state space model approach with classic ERP features, to provide a comparison with previous works?

10.   Did the authors perform any hyperparameter optimization for their models? Which criteria did they consider? Grid, Random, Bayesian search, other? Which kind of kernel did they consider for SMV?

11.   Discussion section is missing, as well as any comparison with previous approaches or machine-learning study on Go/No Go task. Please expand the Conclusions section adding more details and implications about the use of the proposed approach. 

Author Response

Below are the answers to each question. The changes have been made in the paper.

  1. In results section authors compared their models considering parametrization (does it means after state space model step?) and without parametrization using 4 derivations. Why did they not present the results regarding 62 electrodes without parametrization? It is a crucial point to assess the quality of the models and methods. Otherwise, it is not clear to me how the authors can report the following statement “Without parameterization, the algorithms had difficulty with signal classification when all 62 scalp electrodes were used in the model.” I’m referring mainly to Table 1 and its related discussion.

Thank you for your comment. Thanks to it, we checked the results again and performed the analysis for the entire signal without parameters. The results turned out to be high, as described in the revised version of the article. The results are described in sections 9. Results and 10. Conclusion.

  1. Moreover, it seems that the models performed better without parametrization. Therefore, did it mean that the state space model is inefficient or not helpful? In other words, why anyone should consider using this approach if it is not able to improve performance?

There is only one case where better results were obtained for a parameterized signal (LR algorithm for 62 electrodes). However, the statement that the state space model is inefficient or not helpful is not correct. After applying the state space modeling, the results are still high and we have fewer input parameters.

  1. Introduction is too short, background and previous studies regarding ERP analysis and machine-learning analysis in Go/No task have been poorly explained. Moreover, the rationale of the study is not well presented.Authors should better clarify why it could necessary the use of machine-learning approaches in this field, as well as why state space model should be preferred to other techniques.

We made changes in the section 1 Introduction.

  1. Please provide the IRB approval code.

The IRB approval code (IRB approval # 2016-226-NSU) was added to Materials and Methods (Section 2)

  1. Was enough the latency of 600ms to discriminate the ERP components from the Go/NoGo task and the ones from the auditory stimulus? 

Our focus was on the ERP to the second stimulus in each trial (i.e., the stimulus that required a response on Go trials or response withholding on NoGo trials. Thus, while there would have been overlap between the ERP to the tone burst and the ERP to the first stimulus in each trial, the ERP to second stimulus was free of overlap because the second stimulus was presented 1200 ms after the first stimulus. 

Also considering the sampling frequency used. Authors should add some references to justify this choice.

The 500 Hz sampling rate exceeds the Nyquist frequency and is commonly used in cognitive neuroscience research. A reference (Jing and Takigawa, 2000) was added under the EEG recording and processing subsection

  1. “Eyeblinks exceeding ±75 µV were corrected using the covariance method.” Please add a reference and further details regarding the use of this artifact correction method.

A citation was added (Semlitsch, Anderer, Schuster, and Presslich, 1986) along with the following explanation of the covariance method to the EEG recording and processing section. The covariance analysis is performed between the eye artifact channel and each EEG channel. Linear transmission coefficients, similar to beta weights, are computed. Based on the weights, a proportion of the voltage is subtracted from each data point.

  1. Regarding the demographic information, did the authors consider the gender as confounding factor in their study?

No, the authors did not consider gender as a confounding factor in their study.

  1. “Here we have taken EEG data from a single electrode and conducted a system identification exercise on the data.” Which electrode did the authors consider? Did the parameters change according to the electrode selected?” “Several tests showed that not all electrodes had the same system order properties as the example above.” How many tests did the author performed? Did they have any supplementary materials to justify their choice of the system order?

We selected electrode 20 as an example and fitted a state-space model to it. The system order was between 17 and 21. We chose n = 20 and the mean squared error (MSE) was in the order of 10-7. Not all electrodes showed a constant system order throughout. However, the average system order was about 20. For each electrode we conducted an optimization of system order versus MSE. We decided to take n = 20 as an average system order and either truncate or zero pad the transfer function parameters accordingly. This was a result of the decaying behavior of the parameters in the transfer function as the system order increased. So, we started with a minimum system order of n = 17 and calculated MSE. We then incremented the system order to n = 18 and calculated the new MSE. It the new MSE improved, we kept increasing the system order by one, thus trying to bound the MSE to a minimum. So in essence, we sort of found the optimal system order for each electrode. Since we computed the transfer function parameters, we either truncated the parameters to n = 20 or zero padded them in cases where n < 20. The singular value plot versus system order is a common tool used in state-space modeling for determining the system order.

  1. Why did author not compare their state space model approach with classic ERP features, to provide a comparison with previous works?

We added in the Introduction in paper.

  1. Did the authors perform any hyperparameter optimization for their models? Which criteria did they consider? Grid, Random, Bayesian search, other? Which kind of kernel did they consider for SMV? 

Yes, grid search was used. In SVM the rbf kernel is used.

  1. Discussion section is missing, as well as any comparison with previous approaches or machine-learning study on Go/No Go task. Please expand the Conclusions section adding more details and implications about the use of the proposed approach. 

We added in the Conclusion in paper.

Reviewer 2 Report

Comments and Suggestions for Authors

In this study, machine learning methods, including support vector machine (SVM), logistic regression (LR), and artificial deep neural network (DNN), were utilized to classify event-related brain potentials (ERPs) recorded from healthy young adults during a Go/NoGo task, a cognitive inhibition test. The ERPs were first processed using a dynamic state space model, reducing the dimensionality of the data by extracting transfer function parameters. Then the three machine learning approaches were examined to classify the electrical signals associated with different trial types. The study shows the effectiveness of dimensionality reduction in enhancing signal classification and the potential of machine learning in analyzing neural activity.

1. I would suggest providing an overall flowchart of the analysis procedure to help readers better understand the whole pipeline.

2. It would be helpful to provide interpretation of important features for classifying ERP signals. I also believe that it will make this paper stronger if the authors present some insightful implications based on their experimental outcomes.

3. There are many relevant studies in terms of both feature selection and classifier training for EEG classification, but were not mentioned in this paper. The authors need to further elaborate the introduction or review of the existing literatures, for example: Sparse Bayesian learning for end-to-end EEG decoding; A survey on deep learning based non-invasive brain signals: recent advances and new frontiers.

4. In a separate paragraph it is required to provide some remarks to further discuss the proposed methods, for example, what are the main advantages and limitations in comparison with existing methods?

5. The authors may briefly discuss the potential limitations of the proposed method and what are the future research directions of this study. How other researchers can work on your study to continue this line of research?

Author Response

Below are the answers to each question. The changes have been made in the paper.

Q1 I would suggest providing an overall flowchart of the analysis procedure to help readers better understand the whole pipeline.

An overall flowchart of the analysis procedure was added - lines 294-311 and Figure 6 (section 8. Results).

Q2 It would be helpful to provide interpretation of important features for classifying ERP signals. I also believe that it will make this paper stronger if the authors present some insightful implications based on their experimental outcomes.

We added in lines 141-155, 397-406

Q3 There are many relevant studies in terms of both feature selection and classifier training for EEG classification, but were not mentioned in this paper. The authors need to further elaborate the introduction or review of the existing literatures, for example: Sparse Bayesian learning for end-to-end EEG decoding; A survey on deep learning based non-invasive brain signals: recent advances and new frontiers.

We changed the Introduction.

Q4 In a separate paragraph it is required to provide some remarks to further discuss the proposed methods, for example, what are the main advantages and limitations in comparison with existing methods?

We added in the Conclusion (lines 397-406,444-453).

Q5 The authors may briefly discuss the potential limitations of the proposed method and what are the future research directions of this study. How other researchers can work on your study to continue this line of research?

We added in the Conclusion lines 397-406,444-453.

Reviewer 3 Report

Comments and Suggestions for Authors

The paper compares three machine learning algorithms for ERP classification in a Go/NoGo task. However, the rationale for algorithm selection lacks clarity, and the paper lacks novelty in method development or neuroscience findings. The literature review on state-of-the-art ML algorithms for ERP detection in the Go/NoGo task is insufficient. The introduction primarily focuses on method explanation rather than conveying the study's intentions, goals, and novelty. The study's only significant finding is the utility of dimensionality reduction before classification, a concept well-established in classical ML studies. The overarching goals and intentions of this simple classification task involving over 260 subjects are not clearly defined in the current version. Some suggestions for improvement include:

1)      Given that 268 participants were included, it's implied that this study might be part of a larger investigation. Could you provide a reference or information about the main study from which this research is derived or connected?

2)      The authors should present temporal, spectral, and spatial features of the Go/NoGo task in a plot to demonstrate their differences.

3)      The paper lacks thorough discussion to interpret the results; additional analysis and insights are needed for a comprehensive understanding.

Comments on the Quality of English Language

Minor editing of English language is required

Author Response

Below are the answers to each question. The changes have been made in the paper.

Q1      Given that 268 participants were included, it's implied that this study might be part of a larger investigation. Could you provide a reference or information about the main study from which this research is derived or connected?

Lines 86-93. We used a subset of the data collected as part of a larger study funded by the National Science Foundation (Award 1632377) awarded to Mercedes Fernandez. The project evaluated inhibitory control of attention in a linguistically diverse population. We explain that the data were collected as part of a larger study and we cite the study (Fernandez et al., 2023) at the start of the Material and Methods section. 

Q2     The authors should present temporal, spectral, and spatial features of the Go/NoGo task in a plot to demonstrate their differences.

There are many dimensions to this problem. On the one hand there are 64 electrodes distributed spatially, along with 250 selected data points. This could have led to a nice space-time autocorrelation analysis using the spatial structure of the electrodes. However, we didn’t have the right computational tools to do this and it would have taken an enormous amount of time to program it. Likewise, we were not interested in building models for individual subjects, but rather a group of subjects. Thus, we looked at average behavior. That is, we took the average of all electrodes, then we computed the auto and cross correlograms for the Go and NoGo trials. There are significant differences in the autocorrelation pattern between the Go and NoGo trials. For instance, one can observe a periodic behavior in the autocorrelations of the NoGo trials, which is not evident in the Go trials. Likewise, there are an equal amount of positive and negative cross autocorrelations, but more positive than negative autocorrelations in the individual Go and NoGo trials. On the other hand, the average auto and cross autocorrelations tend to be positive. Similarly, if we look at the averaged signals, the spectrum of the Go has a peak at around 300, whereas for the NoGo trials, the spectrum tends to be flat. Finally, the average DC power for the Go trials is smaller than that of the NoGo trials. Therefore, there is sufficient evidence to support that there are significant differences in the dynamic behavior of the Go and NoGo trial data.

Q3      The paper lacks thorough discussion to interpret the results; additional analysis and insights are needed for a comprehensive understanding.

We added in the Conclusion lines   397-406,444-453.

Round 2

Reviewer 1 Report

Comments and Suggestions for Authors

The revised manuscript by Bryniarska et al. appears  improved in terms of readability and content. The authors answered most of my required accurately. However, there are still some dubious points that need to be clarified to consider the manuscript worth to be published.

Below I reported my main criticisms:

Point 1: “There are several techniques used to reduce the dimensionality of EEG data: Linear Discriminant Analysis (LDA), Principal Component Analysis (PCA), and Independent Component Analysis (ICA) [20].

Discrete Wavelet Transform is also often used for this purpose.” Did the authors compare their methods with some of these techniques? This could be a valuable point for the manuscript to assess the quality of the proposed method.

Point 2: The power spectrum in Figure 2b seems in normalized frequencies. Authors should show the power spectrum in the typical frequency range for EEG analysis. In Figure 1a the y-axis should be Autocorrelation not Correlogram.

Point 3: Figure 3 should be better explained in the manuscript or revised. I suppose that time scales were in milliseconds not in seconds. Also, why is the frequency on the WVD plots expressed in MHz and not in Hz?

Finally, please add the unit of measurement in the following statement: “Similarly, if we look at the averaged signals, the spectrum of the Go trials has a peak at around 300”.

Point 4: “Therefore, there is sufficient evidence to support that there are significant differences in the dynamic behavior of the Go and NoGo trial data.” Authors should add statistical analysis to support this statement.

Point 5: The authors replied to my previous point as follows: “There is only one case where better results were obtained for a parameterized signal (LR algorithm for 62 electrodes). However, the statement that the state space model is inefficient or not helpful is not correct. After applying the state space modelling, the results are still high, and we have fewer input parameters.” I agree that performance remained quite high, but still lower than the cases without the parametrization. They should add more details in Table 1 or add additional results in the Results section to support the benefits of using the state space model versus not using it. As an example, they could report in another table the performance of other methods for dimensionality reduction (e.g. PCA, LDA, DWT or others). As well as they could test the differences

between models’ results with statistical analysis (e.g. parametrization vs. without parametrization, 4/62electrodes). Furthermore, they could report how performance varied according to the model order. I’m aware

that they have already reported that parametrization reduced the number of parameters (but in the sections related to the methods). However, it should also be reiterated by reporting quantitative information in the Results section.

Point 6: “In the case of systems where the number of processed signal parameters by the ML algorithm is important, for example due to the signal processing time, the use of this method is justified.” Authors did not report any details regarding the time required for State Space Modeling step nor in training/validation or test phases. Is this operation compatible with real-time processing or contexts where a low signal processing time is required? Nor they reported the average time required for the classification task in function of the classifier

evaluated. It may support the claim that their method could be used to reduce the pre-processing and/or classification time. The same observation can be made for the following statement: “This approach may be useful in real-time systems where fast classification is important even if it is less accurate”.

Point 7: Figures 9 and 10 could be removed.

Point 8: “Compared to existing methods, the use of state space modeling on preprocessed data used in ML algorithms makes it possible to reduce the size of the input data.” Authors should compare their method with other approaches.

Point 9: “The singular value plot versus system order is a common tool used in state-space modeling for determining the system order.” Please add a reference to support this paragraph.

Point 10: “For instance, it may be that ML algorithms may help to distinguish psychogenic non-epileptic seizures (PNES) from epilepsy, especially in cases where only short-term EEG data are available”. This

paragraph, in my opinion, is quite out of context. Rather authors should add comments or references related to ML in ERP analysis or in Go/No task field. Possibly suggesting how their method may be considered as a valid alternative of such approaches. Moreover, they could add a short paragraph regarding the current limits and pitfalls of their method.

Point 11: “Furthermore, it is important to weigh the trade-offs between size of the data matrices and the number of parameters, where a parsimonious model (i.e., a model with a minimum number of parameters) is always preferred.” I can agree with this comment, however for deep-learning approaches (still considered as a branch of ML) it may not always be true.

Point 12: Could the authors report in the manuscript which 4 electrodes were selected? Was electrode 20 included in the 4-electrodes experiments? In other words, did the author check how the model order selection influenced the models’ performance? Furthermore, If the 4-electrodes were spatially close, did they check any volume conduction effect on them?

Comments on the Quality of English Language

minor editing required

Author Response

Point 1: “There are several techniques used to reduce the dimensionality of EEG data: Linear Discriminant Analysis (LDA), Principal Component Analysis (PCA), and Independent Component Analysis (ICA) [20]. Discrete Wavelet Transform is also often used for this purpose.” Did the authors compare their methods with some of these techniques? This could be a valuable point for the manuscript to assess the quality of the proposed method.

We have added PCA analysis as a dimensionality reduction technique. We have also used LDA as an ML methods.

Point 2: The power spectrum in Figure 2b seems in normalized frequencies. Authors should show the power spectrum in the typical frequency range for EEG analysis. In Figure 1a the y-axis should be Autocorrelation not Correlogram.

All along we had a pretty good idea that using a dynamic model to reduce the data would work, but we lacked the details and failed to convey the message. In this new version, we re-did the data analysis, considered six ML models, and added many pictures and graphs showing all the details. I believe the material on power spectrum and autocorrelations are no longer necessary. 

We replaced spectral figures with ERP figure 6, which we believe will be more useful to the reader since the paper relates directly to ERP. Figure 6 shows the entire ERP signal (i.e., the 471 data points) which we used to test the ML algorithms. This figure also shows the truncated portion of the ERP from which parameters were extracted. These parameters were used as substitutes for the original data (i.e., the 471 data points) and the ML algorithms were trained on the substitute data. This approach allowed the direct assessment of the effectiveness of dimensionality reduction to EEG analyses.

Point 3: Figure 3 should be better explained in the manuscript or revised. I suppose that time scales were in milliseconds not in seconds. Also, why is the frequency on the WVD plots expressed in MHz and not in Hz? Finally, please add the unit of measurement in the following statement: “Similarly, if we look at the averaged signals, the spectrum of the Go trials has a peak at around 300”.

We removed spectral information, including Figure 3, as we do not believe it added meaningful information to the updated version of the manuscript.   

Point 4: “Therefore, there is sufficient evidence to support that there are significant differences in the dynamic behavior of the Go and NoGo trial data.” Authors should add statistical analysis to support this statement.

This was part of the section on spectral differences which we removed from the manuscript as stated above. However, in the manuscript, we cite authors (Nieuwenhuis et al., 2003, Fernandez et al., 2013; 2014) who have published manuscripts on the differences between the Go and NoGo ERP.

Point 5: The authors replied to my previous point as follows: “There is only one case where better results were obtained for a parameterized signal (LR algorithm for 62 electrodes). However, the statement that the state space model is inefficient or not helpful is not correct. After applying the state space modelling, the results are still high, and we have fewer input parameters.” I agree that performance remained quite high, but still lower than the cases without the parametrization. They should add more details in Table 1 or add additional results in the Results section to support the benefits of using the state space model versus not using it. 

As an example, they could report in another table the performance of other methods for dimensionality reduction (e.g. PCA, LDA, DWT or others). As well as they could test the differences between models’ results with statistical analysis (e.g. parametrization vs. without parametrization, 4/62electrodes). Furthermore, they could report how performance varied according to the model order. I’m aware that they have already reported that parametrization reduced the number of parameters (but in the sections related to the methods). However, it should also be reiterated by reporting quantitative information in the Results section.

Thank you very much for your input regarding this portion of our manuscript. In response to your feedback, we employed PCA as part of dimensionality reduction and tested the parameterized (substitute) data on 6 ML algorithms. This approach was very successful. Not only the results showed equally accurate to using the raw data, the use of state-space modeling reduced the data by 83%. We have added two pictures showing how the two procedures process the data.

Point 6: “In the case of systems where the number of processed signal parameters by the ML algorithm is important, for example due to the signal processing time, the use of this method 2 is justified.” Authors did not report any details regarding the time required for State Space Modeling step nor in training/validation or test phases. Is this operation compatible with real-time processing or contexts where a low signal processing time is required? Nor they reported the average time required for the classification task in function of the classifier evaluated. It may support the claim that their method could be used to reduce the pre-processing and/or classification time. The same observation can be made for the following statement: “This approach may be useful in real-time systems where fast classification is important even if it is less accurate”.

In the state space calculations, we fit 62 models (1 per electrode), then we re-arrange the data as a 510 x 4,960 data matrix, much smaller than 510 x 29,202 (full data case). Then we run all six ML models. This entire procedure takes about 10-15 minutes because we are running the models iteratively, as a function of number of principal components. A formal time analyses of the operation of the ML algorithms or parameterization have not been done. If we knew how many principal components would be necessary in advance so that the analysis doesn’t have to be done iteratively, the procedure would run very fast. Since, to the author’s knowledge, this approach is new, a formal time analysis can be done at a future time. In the paper we highlight each procedure and supported with accurate results.

Point 7: Figures 9 and 10 could be removed.

The figures were removed.

Point 8: “Compared to existing methods, the use of state space modeling on preprocessed data used in ML algorithms makes it possible to reduce the size of the input data.” Authors should compare their method with other approaches.

We compared 6 ML algorithms on several metrics before and after applying state space modeling and PCA feature selection. Results revealed that the metrics were equally strong, with one exception, whether the entire ERP signal was used or the reduced parameterized substitute data was used.   

Point 9: “The singular value plot versus system order is a common tool used in state-space modeling for determining the system order.” Please add a reference to support this paragraph.

We added a reference. In fact, many of the references for this topic cover such idea.

Point 10: “For instance, it may be that ML algorithms may help to distinguish psychogenic non-epileptic seizures (PNES) from epilepsy, especially in cases where only short-term EEG data are available”. This paragraph, in my opinion, is quite out of context. Rather authors should add comments or references related to ML in ERP analysis or in Go/No task field. Possibly suggesting how their method may be considered as a valid alternative of such approaches. Moreover, they could add a short paragraph regarding the current limits and pitfalls of their method.

Thank you for the suggestion, we replaced the example with the following. “For instance, in early or pre-clinical cases associated with deficient inhibition, such as ADHD and Parkinson's Disease, ML algorithms may assist with early detection and diagnosis since research reveals smaller NoGo N2 ERP amplitude in patients compared to controls [Smith2004, Wu2019]. In preclinical cases, ML algorithms may detect small changes in the N2 ERP signal that may be missed by visual inspection alone.

Point 11: “Furthermore, it is important to weigh the trade-offs between size of the data matrices and the number of parameters, where a parsimonious model (i.e., a model with a minimum number of parameters) is always preferred.” I can agree with this comment, however for deep-learning approaches (still considered as a branch of ML) it may not always be true.

Thank you for your input. Please note that we removed from the current version of the manuscript the DNN algorithm we had included in our original submission. The described research constitutes a proposal that can be applied to various studies on EEG signals. As a result of our research, it turned out that the use of State Space Modeling reduced the dimensionality of the input data and yielded equally strong results. Further research is certainly needed on the validity of using this method.

Point 12: Could the authors report in the manuscript which 4 electrodes were selected? Was electrode 20 included in the 4-electrodes experiments? In other words, did the author check how the model order selection influenced the models’ performance? Furthermore, If the 4-electrodes were spatially close, did they check any volume conduction effect on them?

In the current version of the manuscript, we conducted all analyses on all 62 electrodes. We removed all prior results involving only 4 electrodes. Regarding volume conduction, EEG signals have low spatial resolution because of this electrical spread. The advantage of ERP recordings is its temporal resolution. In general, ERPs are described as either a positive- or negative- going wave, the timing of the maximum amplitude of the target wave and the spatial location where the signal reaches maximum amplitude. Based on prior findings, we expected the strongest signal to be on the right frontal-central electrodes sites.

Reviewer 3 Report

Comments and Suggestions for Authors

After a thorough review, I must express my concerns that the paper still lacks novelty in both the fields of machine learning and neuroscience. Additionally, the new signal processing introduced is scientifically weak and does not contribute substantively to the overall quality of the paper.

Author Response

After a thorough review, I must express my concerns that the paper still lacks novelty in both the fields of machine learning and neuroscience. Additionally, the new signal processing introduced is scientifically weak and does not contribute substantively to the overall quality of the paper.

We thank you for your feedback. We are confident that you will agree that the revised manuscript highlights the novel aspects of our approach. We used state-space modeling and PCA substitute and reduced the size of original data by 83%. Our findings reveal that ML algorithms yield equally high metric-wise (accuracy, sensitivity, etc) with the original large dataset as with the much-reduced parameterized data set. To our knowledge there is no published work that compares ML results involving the original ERP data versus a reduced parameterized data set. We believe that the state-space methodology we proposed is designed with ERP signals in mind, since these are impulse response type signals. Furthermore, the class of subspace based system identification known as N4SID, are among the best methods for fitting dynamic models to data. The MSE values were on the order of 10-7, so the fitted models were of very high quality. The results confirm our initial hypothesis that the reduced data set would not degrade the quality (accuracy) of the results.

Round 3

Reviewer 3 Report

Comments and Suggestions for Authors

The paper has improved substantially. The quality of some figures such as Figure 3 , Figure 6 is low and must be improved.